# Defending Adversarial Attacks against DNN Image Classification Models by a Noise-Fusion Method

**Lin Shi \*, Teyi Liao and Jianfeng He**

Department of Computer Science, Faculty of Information Engineering and Automation, Kunming University of Science and Technology, Kunming 650500, China; 20202204264@stu.kust.edu.cn (T.L.); jfenghe@kust.edu.cn (J.H.)
* Correspondence: lin.shi@kust.edu.cn

**Abstract:** Adversarial attacks deceive deep neural network models by adding imperceptibly small but well-designed attack data to the model input. Those attacks cause serious problems. Various defense methods have been provided to defend against those attacks by: (1) providing adversarial training according to specific attacks; (2) denoising the input data; (3) preprocessing the input data; and (4) adding noise to various layers of models. Here we provide a simple but effective Noise-Fusion Method (NFM) to defend adversarial attacks against DNN image classification models. Without knowing any details about attacks or models, NFM not only adds noise to the model input at run time, but also to the training data at training time. Two $l_\infty$-attacks, the Fast Gradient Signed Method (FGSM) and the Projected Gradient Descent (PGD), and one $l_1$-attack, the Sparse L1 Descent (SLD), are applied to evaluate defense effects of the NFM on various deep neural network models which used MNIST and CIFAR-10 datasets. Various amplitude noises with different statistical distribution are applied to show the defense effects of the NFM in different noise. The NFM also compares with an adversarial training method on MNIST and CIFAR-10 datasets. Results show that adding noise to the input images and the training images not only defends against all three adversarial attacks but also improves robustness of corresponding models. The results indicate possibly generalized defense effects of the NFM which can extend to other adversarial attacks. It also shows potential application of the NFM to models not only with image input but also with voice or audio input.

**Keywords:** adversarial attack; defense; noise; deep neural network

## 1. Introduction

With the development of deep neural network (DNN) models and the advent of high-performance hardware, deep learning has made remarkable progress in traditional fields such as image classification, speech recognition and language conversion. However, adversarial attacks deceive DNN models by adding imperceptibly small but well-designed attack data to the model input which cause serious problems. Attacking data injects into the DNN model input in the form of image [1–4] and audio [5,6]. The impact of attacks originates from the imperceptible small changes to the model input, whereas the model is treated by the tiny "noise" while human sensation could not even notice the changes.

There are two types of adversarial attacks: the white-box attack and the black-box attack. The white-box attack requires complete knowledge of models, e.g., structure of models and the specific parameters of each layer. The attacks control the input of the models or modify the input data. Most adversarial attacks are classified as white-box attacks. Common white-box attacks include the Fast Gradient Signed Method (FGSM) [7], the Projected Gradient Descent (PGD) [8], the Sparse L1 Descent (SLD) [9], the C and W [10], the DeepFool [11], and adversarial attack methods based on generative adversarial network frameworks [12].

The black-box attack treats the target model as a black box. Without knowing the internal details of the model, the attacks only control the input of the models. There are

few black-box attack methods, including the Substitute Blackbox Attack [13] and the One Pixel Attack [14].

Current adversarial defense methods defend against attacks in four ways as described below.

The first way is to provide corresponding adversarial training of neural network models according to specific attacks, e.g., defensive distillation [15] methods based on adversarial training [16] using thermometer-encoding methods to aid adversarial training [17] and multi-level JPEG compression of images [18].

The second way is to denoise the input data. Handling an adversarial attack as a special kind of noise is the key aspect of denoising methods. For example, wavelet transform-based methods [19,20] were used as a defense method; autoencoders-based methods [21] were used as a defense; data transformation methods [22] were used as a defense method; median filtering [23] was employed as a defense method.

The third way is to preprocess the input data. It tries to preprocess the input data to erase attack data. For example, preprocessing images used in the Erase-and-Restore method [24], using image rotation as a preprocessor [25], and using PCA as a preprocessor [26].

The fourth way is to add noise to the model layers. Gaussian noise injection is performed in each layer of neural network activations or weights to improve the robustness of neural networks to adversarial attacks [27]. The generalization ability of CNNs was improved by adding well-designed noise to the middle layer activations [28]. Noise was injected into hidden layers in a hierarchical manner to improve the robustness of neural networks [29]. Those methods required a sufficient understanding of the relevant neural network models to provide a defense effect.

Furthermore, many defenses are complex in design and were breached shortly after being provided. For example, a defensive distillation method with good results was proposed in [15], while it was soon broken down in [10]. The magnet and feature squeezing proposed in [30,31] were broken by the adaptive attack proposed in [32,33]. Defense methods in [13,34–36] proposed methods to make neural networks more robust were still broken down by the research of [10,37].

Here, we provide a simple defense method against adversarial attacks, named the Noise-Fusion Method (NFM). The NFM does not require any knowledge of the details of the models and the attacks. The NFM simply adds noise to the model inputs and to the training data to defend against attacks.

Our method defends against adversarial attacks by adding noise not only to input data but also to training data. Without knowing any details about the relevant neural network models, three types of noise, the Uniform noise, the Gaussian Noise, and the Poisson noise, were applied to the MNIST and the CIFAR-10 datasets and corresponding models to evaluate the effect of the NFM on those three common adversarial attacks.

## 2. Method

We designed two experiments using MNIST and CIFAR-10 datasets and the corresponding deep neural network models. Three adversarial attacks and three kinds of noises were used in the experiments to demonstrate the effectiveness, accuracy and robustness of the method.

### 2.1. Adversarial Attack Methods

The experiment used two $l_\infty$-attacks, the Fast Gradient Signed Method (FGSM) [7] and the Projected Gradient Descent (PGD) [8], and one $l_1$-attack, the Sparse L1 Descent (SLD) [9] to evaluate the effects of the NFM. The $l_\infty$-attack strove to minimize the change of the pixel with the largest change. The $l_1$-attack referred to the process of adversarial attack generation that sought to minimize the $x$ and $x^{adv}$ Euclidean distances while ensuring that the deep neural network was successfully misled. The $x$ represented data that was not under attack; $x^{adv}$ was the maliciously attacked $x$.

The FGSM attack was an algorithm for generating adversarial samples based on the gradients of the models. The FGSM attack calculated the direction of the gradient through

a sign function. The sign function was a function used to find the sign of a value. For input greater than 0, the output was 1; for input less than 0, the output was −1, and for input equal to 0, the output was 0. The influence of the gradient disturbance became bigger and bigger like a snowball to achieve the purpose of attack.

The PGD attack was an iterative attack, which can be regarded as a replication of the FGSM and the K-FGSM (K represents the number of iterations). The FGSM only performed one iteration and took a big step, while the PGD did multiple iterations, taking a small step each time, and each iteration clipped the disturbance to a specified range.

The SLD implemented a variant of the $l_1$-norm's projected gradient descent. The $l_1$-norm case was trickier than the $l_\infty$ and $l_2$ cases covered by the PGD class because the steepest descent direction for the $l_1$-norm was too sparse (it updated one coordinate in the adversarial perturbation at each step). This attack had an additional parameter to control the sparsity of the update step. For moderately sparse updated steps, this attack significantly outperformed the predicted steepest descent method and was more competitive with other attacks against the $l_1$-norm.

The FGSM, PGD and SLD attacks implemented by CleaverHans [38] were used to generate the adversarial attack examples. Noise was added into images at training time and running time, respectively. A sequential step was defined as Algorithm 1 to generate and fuse noise into images. Amplitudes of three statistical distributions of noise (Uniform noise, Gaussian noise and Poisson noise) were set to 0%, 10%, 20%, 30%, 40%, 50%, 60%, 70%, 80%, 90% and 100%, respectively. Training images with and without noises were used to train the model alternately.

---

**Algorithm 1** Adding Noise

---

**Input** batch data $(x, y)$
**Output** predicted value $\hat{y}$
**Training phase:**
**1: Repeat**
**2:**   (1) Add noise to the data by: $f(x_{noise}) = x \cdot (1 - amp) + noise \cdot amp, \ amp \in [0, 1]$
**3:**   (2) Calculate losses and noise gradients.
**4:**   (3) Update network parameters.
**5: until** training finishes
**Defensive phase:**
**1:**   Fusion of adversarial sample data with noise.
**2:**   Added it and the image data to the neural network.
**3:**   Given testing image $x$, initialize $p = (0, 0, \ldots, 0)$
**4:**   Forward propagation to calculate probability output $p = f_n(\omega, x)$
**5:**   Update $p$
**6:**   **Predict** $y = argmax\, p$

---

### 2.2. Defense Algorithms

We used three kinds of noise, Gaussian noise, Poisson noise, and Uniform noise, to evaluate the defense effect of the NFM. Uniform noise was generated from a set of random numbers drawn from a uniform distribution in the interval [0,1]. Gaussian noise referred to a class of noise in which the probability density function followed a Gaussian distribution (a normal distribution). The probability density function of Gaussian noise followed a standard normal distribution in which the expectation was 0, and variance was 1. Poisson noise had a Poisson distribution.

We fused the noise with the input data by Equation (1). Gaussian noise, Poisson noise, and Uniform noise were generated using corresponding distributions. The variable '*amp*' indicated the amplitude of the noise. The variable '*noise*' indicated the noise matrix which had the same size as the image matrix.

$$f(x_{noise}) = x \cdot (1 - amp) + noise \cdot amp, \qquad amp \in [0, 1] \tag{1}$$

Figure 1 shows example images from the MNIST and CIFAR-10 datasets with different amplitudes of noise. For the MNIST and CIFAR-10 datasets, the amplitudes of random noise were 0%, 10%, 20%, 30%, 40%, 50%, 60%, 70%, 80%, 90% and 100%, respectively. On this scale, 0% meant no noise and 100% meant no signal.

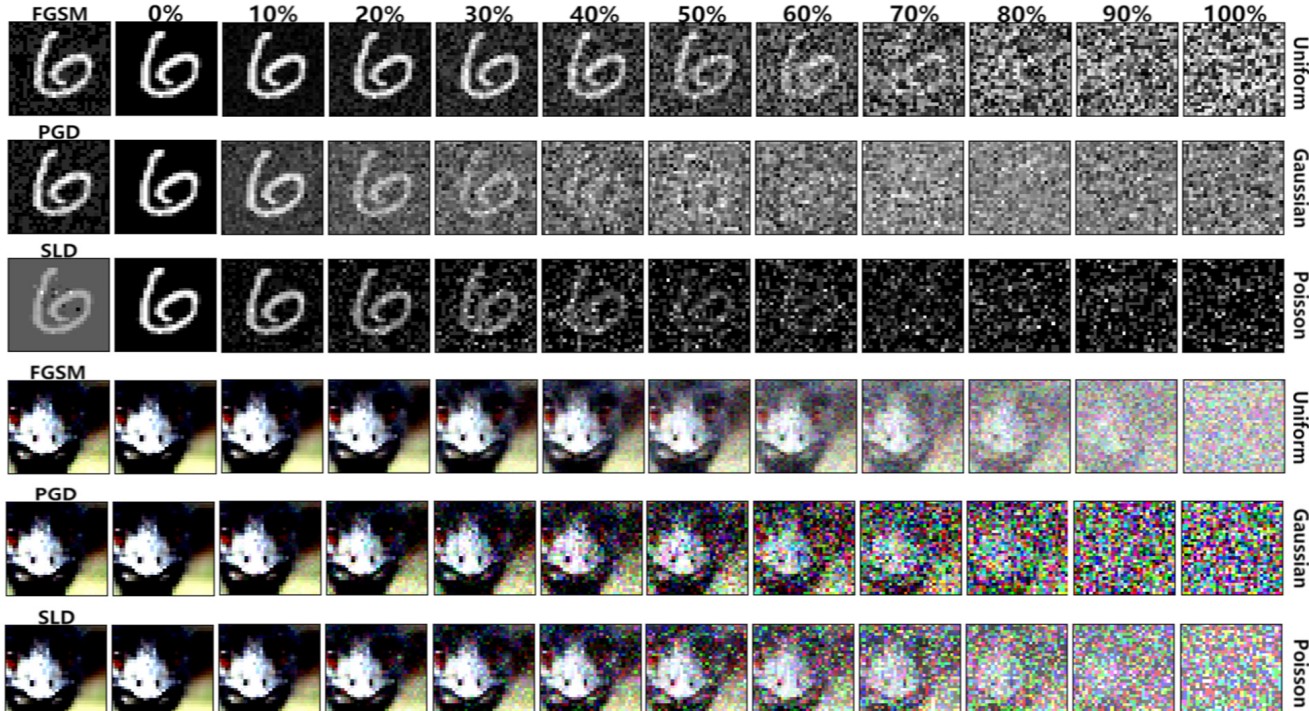

**Figure 1.** Example images with various amplitude noises. Rows 1, 2 and 3 were for MNIST, and rows 4, 5 and 6 were for CIFAR-10. The first column images were images with the FGSM, PGD and SLD attacks, respectively. The images in rows 1, 2, 3 or 4, 5, 6 were images with Uniform noise, Gaussian Noise, and Poisson noise, respectively.

Since the adversarial attack appeared as a small disturbance on the image that was imperceptible to the human eye, our main idea was to blend it with a certain amplitude of noise to drown out the adversarial attack. In addition, to investigate the effect of different types of noise on the defense against the adversarial attack, we used Poisson noise, Gaussian noise, and Uniform noise.

We represented the original neural network as $f(\omega, x)$, where $\omega$ was the weight and $x$ was the input image. The network was represented by $n$ as $f_n(\omega, x)$. Algorithm 1 shows the algorithm steps used in the experiments.

*2.3. Datasets and Related Models*

A convolutional neural network with three convolutional layers and two fully connected layers was used for the MNIST dataset.

An 18-layer ResNet (17 convolutional layers, 1 fully connected layer) was used for the CIFAR-10 dataset with data augmentation (random flipping, shifting, rotation and whitening). The loss function was the common cross-entropy loss function. The optimizer was the SGD and the *lr*, and the momentum was set to 0.01 and 0.9, respectively, in the process of training of corresponding models.

Training images with and without noise were used to train the model alternately in all conditions. The test images also added noises with amplitudes 0%, 10%, 20%, 30%, 40%, 50%, 60%, 70%, 80%, 90% and 100%, respectively, for the MNIST and CIFAR-10 datasets. The accuracy of the test reached 92% after training 135 epochs on the CIFAR-10 dataset with the Resnet network.

Furthermore, we also compared the adversarial training method with the NFM. Adversarial training was a popular adversarial defense method which constructed some adversarial samples and added it to the training dataset to enhance the robustness of the model to defend against adversarial attacks.

## 3. Results

We analyzed the experimental data for the MNIST and CIFAR-10 datasets and the adversarial training method.

### 3.1. Results on the MINST

Without applying the defense method, the accuracy of the model against the FGSM, SLD and PGD attacks were about 33%, 36% and 6%, respectively. With applying the NFM defense method, the maximum accuracy was about 61%, 85% and 76%, respectively.

The following results were obtained from Figure 2.

(1) The accuracy values dropped from over 95% to about 30% under the condition of the FGSM attack without defense, while the accuracy of the model improved to 60% with the NFM defense. Among them, the defense effect of the NFM under the condition of Uniform noise was the best (see Figure 2A–C,J);

(2) The accuracy values dropped from over 95% to below 10% under the condition of the PCD attack without defense, while the accuracy of the model improved to 80% with the NFM defense. Among them, the defense effect of the NFM under the condition of Poisson noise was the best (see Figure 2D–F,K);

(3) The accuracy values dropped from over 95% to below 40% under the condition of the SLD attack without defense, while the accuracy of the model improved to over 80% with the NFM defense. Among them, the defense effect of the NFM under the condition of Poisson noise was the best (see Figure 2G–I,L).

The results obtained from Figures 3 and 4 were as follows:

(1) In the NFM under Gaussian noise conditions, when the input noise amplitude varied from 0% to 40%, the model accuracy was improved in all attack conditions. The NFM under Poisson noise conditions improved model accuracy in all attack conditions with input noise amplitudes from 0% to 80%. In the NFM under Uniform noise conditions, when the input noise amplitude was 60% to 80%, the model accuracy was improved in all attack conditions (see Figure 3);

(2) Under the FGSM attack condition, the model accuracy was improved when adding training noise ranging from 0% to 80% amplitude. Under the PGD attack condition, adding training noise in almost all amplitude intervals improved the model accuracy. Under the SLD attack condition, except adding 100% amplitude training noise, the model accuracy was improved in all cases (see Figure 4);

(3) The NFM under Poisson noise conditions had the best defense effect against all attacks which was obtained at an 80% amplitude Poisson noise condition.

### 3.2. Results on the CIFAR-10

Figure 5 shows the accuracies of test images with various amplitude noises used with the CIFAR-10 dataset. The model was trained using training images with various amplitude noises.

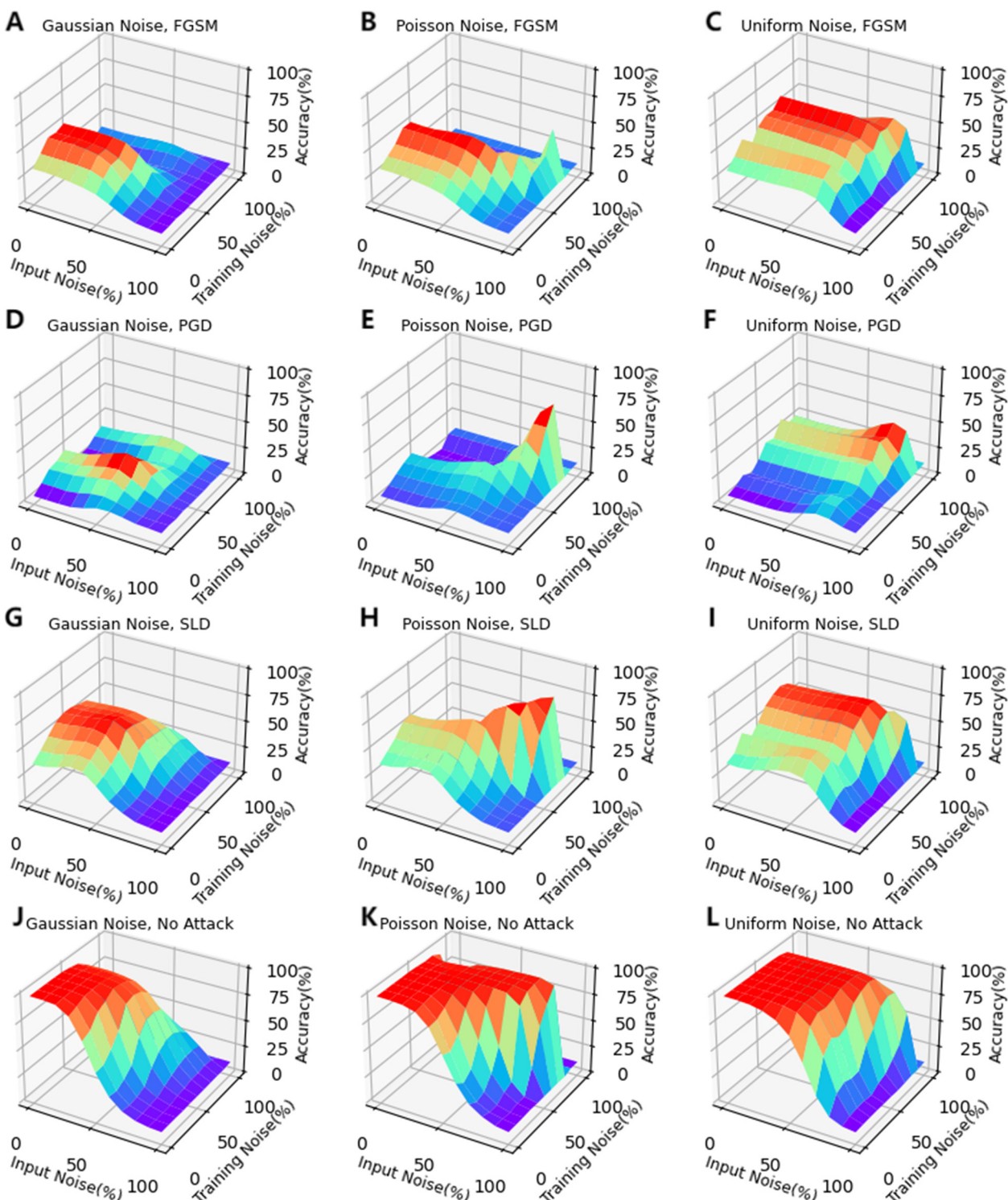

**Figure 2.** Accuracy of the MNIST. The *x* and *y* axes are the amplitudes of the training noise added during training time and the input noise added at running time, respectively. The *z* axis represents the accuracy of the corresponding noise conditions.

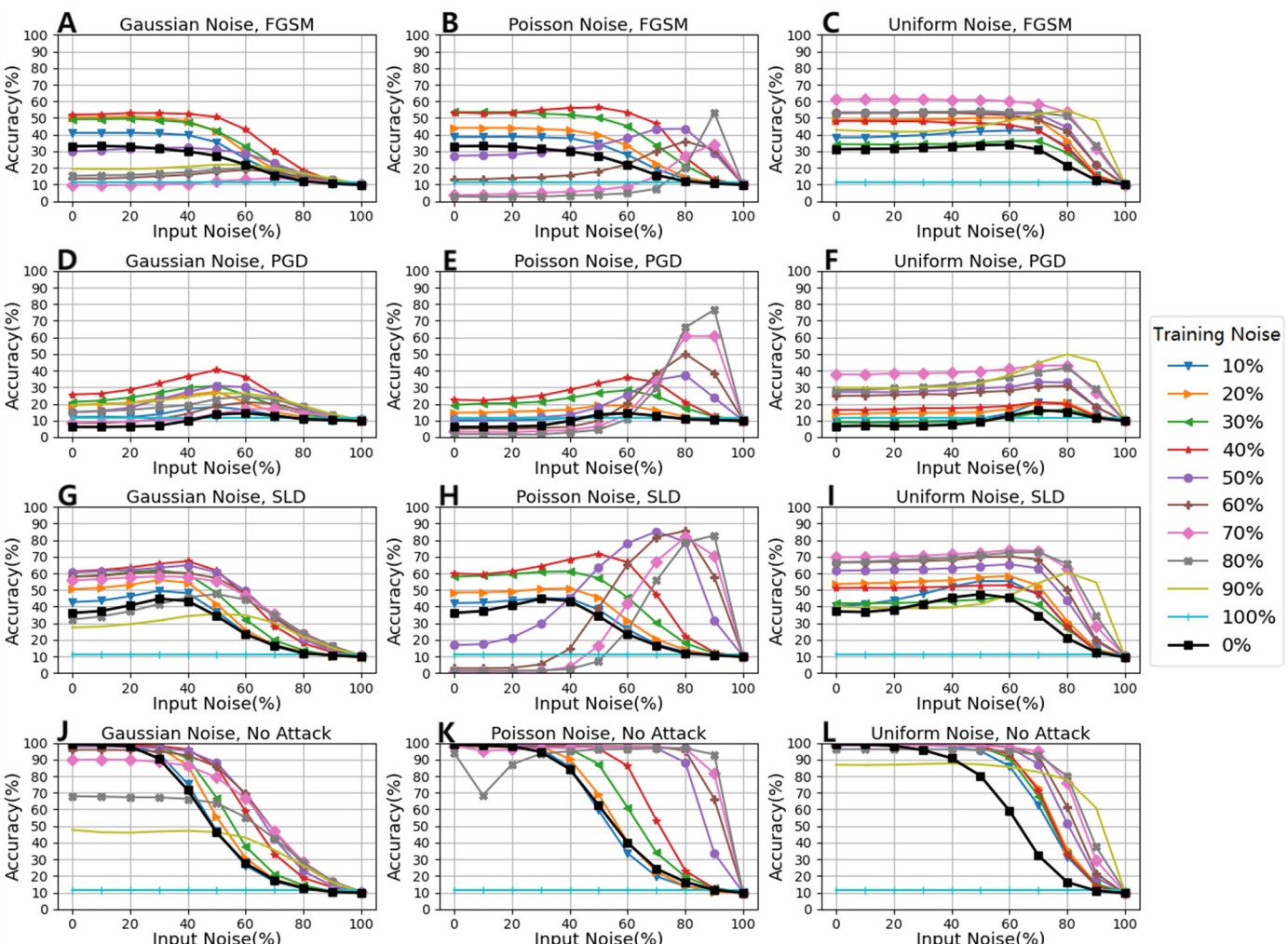

**Figure 3.** Accuracy of the MNIST at input noise dimension. The *x* axis is the input noise amplitude, and the *y* axis is the accuracy. Different color curves with various symbols indicate different amplitudes of training noise.

From Figure 5 we can reach the following conclusions:

(1) The accuracy values dropped from over 90% to below 20% under the condition of the FGSM attack, while the accuracy of the model improved with the NFM to 80%. Among them, the defense effect of the NFM under a condition of Poisson noise and Uniform noise was the best (see Figure 5A–C,J);

(2) The accuracy values dropped from over 90% to below 10% under the condition of the PGD attack, while the accuracy of the model improved to over 80% with the NFM. Among them, the defense effect of the NFM under a condition of Poisson noise and Uniform noise was the best (see Figure 5D–F,K);

(3) The accuracy values dropped from over 90% to about 40% under the condition of the SLD attack, while the accuracy of the model improved to 85% with the NFM. Among them, the defense effect of the NFM under a condition of Poisson noise and Uniform noise was the best (see Figure 5G–I,L).

The results obtained from Figures 6 and 7 were as follows:

(1) On the CIFAR-10 dataset, under the conditions of the FGSM, PGD and SLD attacks, the model accuracy tended to be the highest at a small amplitude noise;

(2) Whether adding input noise or training noise, the model accuracy was significantly improved compared to without adding noise.

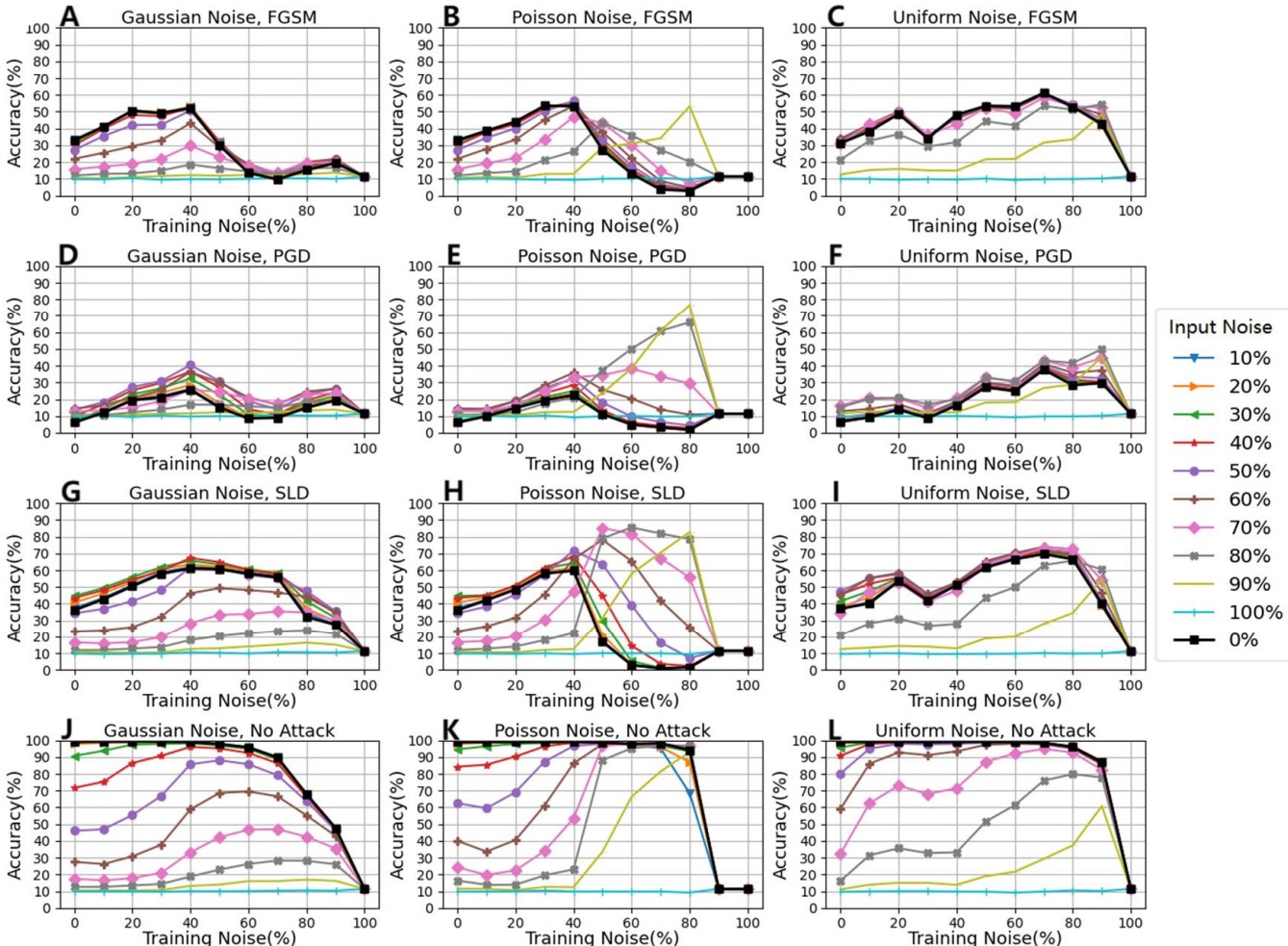

**Figure 4.** Accuracy of the MNIST at the training dimension. The *x* axis is the training noise ampli­tude, and the *y* axis is the accuracy. Different color curves with various symbols indicate different amplitudes of inputting noise.

On the CIFAR-10 dataset, we found that the NFM improved the accuracy in defending against adversarial attacks. The accuracy was 78%, 83% and 89% in defending against the FGSM, PGD and SLD attacks, respectively. We explain the reasons why the NFM was effective in the following discussion section.

### 3.3. Comparison Adversarial Training and the NFM

Table 1 shows the accuracy of adversarial training and the NFM on the MINST and CIFAR-10 datasets, respectively. On the MNIST dataset, adversarial training had higher accuracy values than the NFM under the FGSM and PGD attacks, while adversarial training had lower accuracy values than the NFM under the SLD attack. On the CIFAR-10 dataset, the NFM showed a better defense effect than the adversarial training under all three attacks.

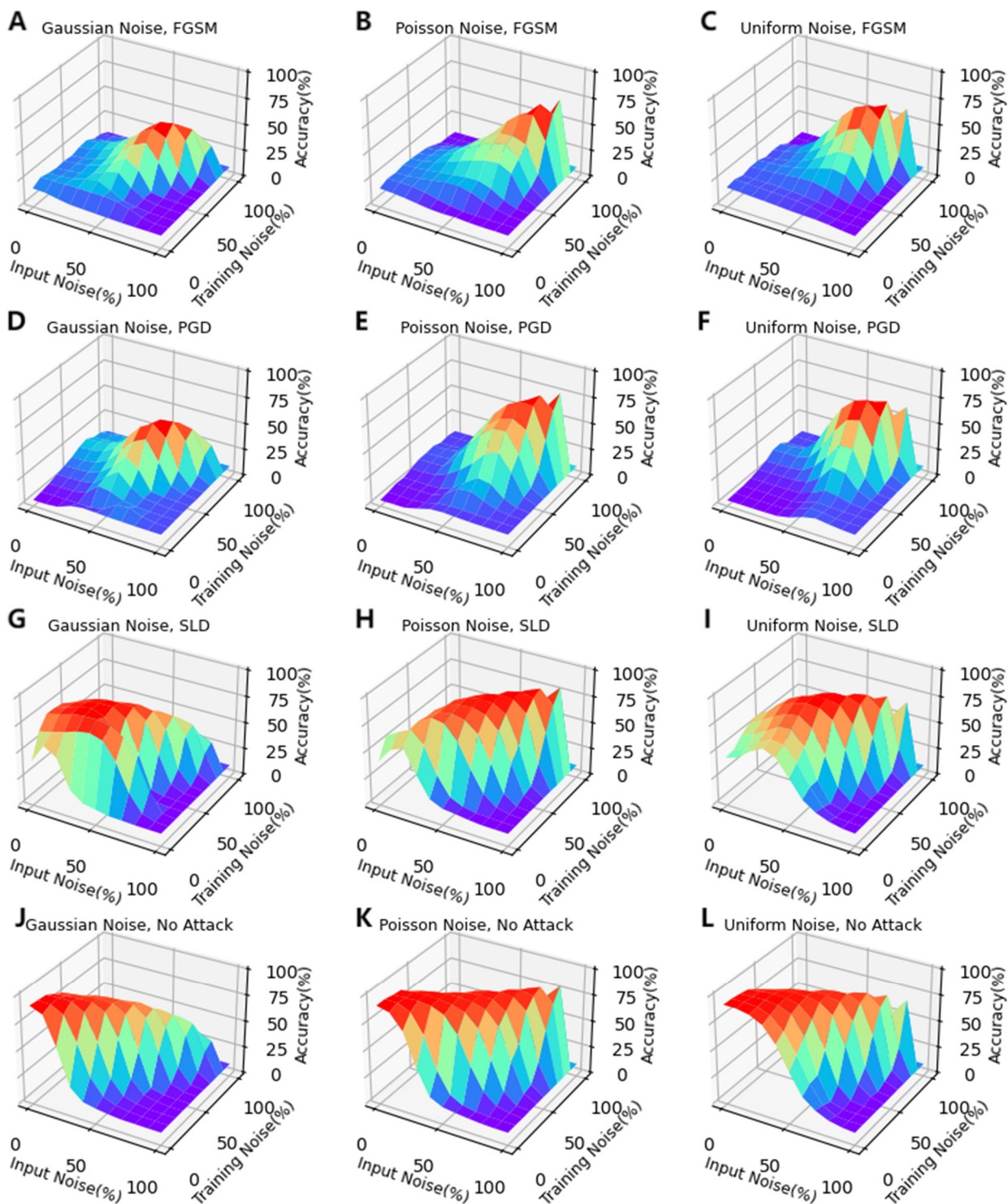

**Figure 5.** Accuracy of the CIFAR-10. The *x* and *y* axes are the amplitudes of the training noise added during training time and the input noise added at running time, respectively. The *z* axis represents the accuracy of corresponding noise conditions.

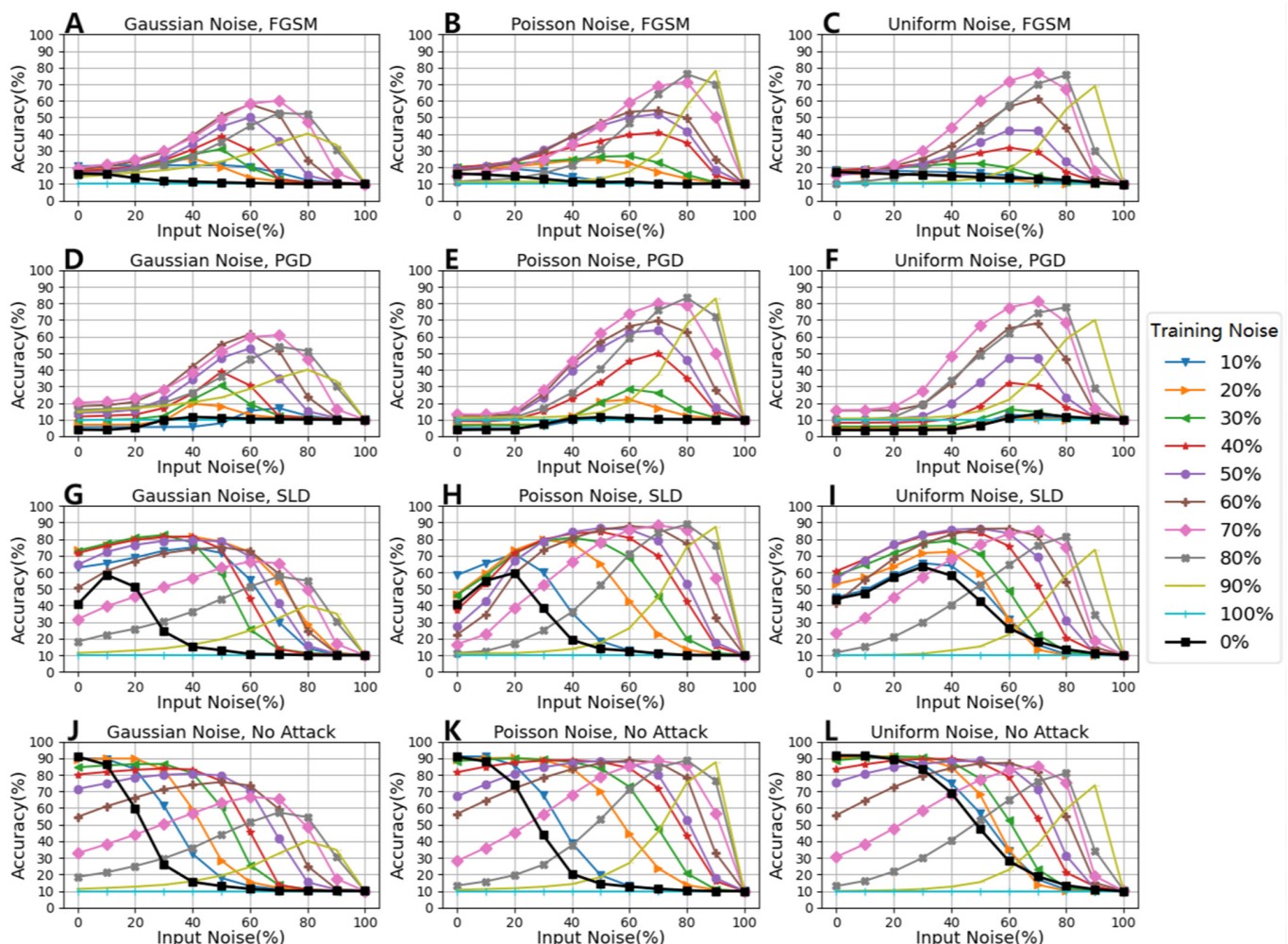

**Figure 6.** Accuracy of the CIFAR-10 at input noise dimension. The *x* axis is the input noise amplitude, and the *y* axis is the accuracy. Different color curves with various symbols indicate different amplitudes of training noise.

**Table 1.** Accuracy of adversarial training and the NFM on the MNIST and CIFAR-10.

| Dataset | Method | FGSM | PGD | SLD |
|---------|--------|------|-----|-----|
| MNIST | Adversarial Training | 95% | 94% | 83% |
| | NFM | 61% | 77% | 86% |
| CIFAR-10 | Adversarial Training | 47% | 46% | 74% |
| | NFM | 78% | 83% | 89% |

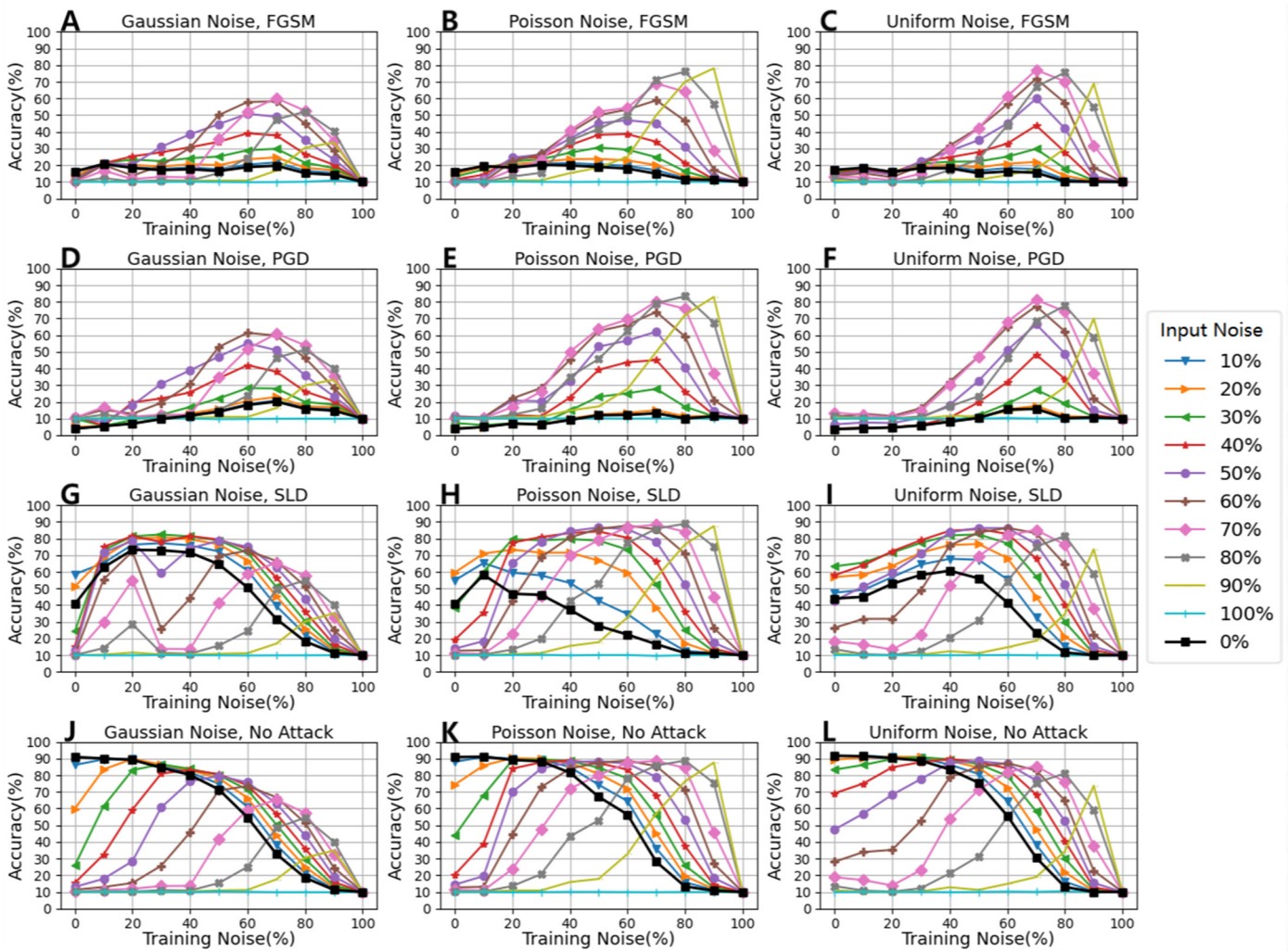

**Figure 7.** Accuracy of the CIFAR-10 at training dimension. The *x* axis is the training noise amplitude, and the *y* axis is the accuracy. Different color curves with various symbols indicated different amplitudes of input noise.

## 4. Discussion

We provided a simple but effective defense method against adversarial attacks. Training time noises not only improved accuracy but also improved robustness of the model. We tested three attacks by adding three types of noise. The results showed that our method was effective in defending against all counterattacks on both the MINST and CIFAR-10 datasets. Adding noises into the model input fused the attacking data within the added noise. Adding noise into training data improved the robustness of the corresponding models.

In experiments using the CIFAR-10 dataset, for $l_\infty$-attacks such as the FGSM and the PGD, our method achieved a significant increase in accuracy when a small noise amplitude was added. For $l_1$-attacks such as the SLD, although the defense effect was not obvious when adding small amplitude training noise and input noise, it still showed good defense effect when gradually increasing the training noise amplitude to 70% and input noise to 90%. When the noise amplitude increased to 100% the neural network failed to classification input images because of complete loss of picture information.

Although the effect of the NFM was lower than the adversarial training method on the MNIST dataset, the effect of the NFM was higher than the adversarial training method on the CIFAR-10 dataset. The adversarial training method improved the accuracy in some cases more than the NFM because of knowing the details of attacks, while the NFM did not know details of the attacks.

### 4.1. Adversarial Defense Effects

The NFM defended against various adversarial attacks on various datasets without knowing any details about the attacks or the models. The NFM already showed the ability to defend against "unknown attacks of unknown models". From the view of the NFM, any attacks were noise. Fusing the noise meant attacks were defended against.

The NFM not only added noise to the training phase to improve the robustness of the network to noise, but also fused the attack data with a variety of noise. Among three noises, Poisson noise tended to have a better defense effect than Gaussian noise and Uniform noise. In this study, we only tested the same independent distribution noises that lacked spatial feature structures. More noise with spatial feature structures will be tested in future works.

In adversarial attack experiments, most of the attacks were gradient-based. For example, noise could have the effect of perturbing gradients in a network, so the NFM had two advantages. First, fusing the attack data into the noise could undoubtedly have a 'drowning' effect against the attack. Second, adding noise to the training network helped the optimization algorithm find a stable convolution filter that was robust to the noisy perturbation input, thus further enhancing the 'drowning' effect.

### 4.2. Accuracy

As shown in panels A–I in Figures 3, 4, 6 and 7, in both the MNIST and the CIFAR-10 datasets, the peak accuracy positions were NOT located at the zero-noise points, which indicates that the NFM improved the accuracy by adding noise. The accuracy values of the zero-noise points (zero input noise and zero training noise points, zero points in 3D figures) indicate the effect without using the NFM, and all other points indicate the effect using the NFM at various noise amplitude conditions.

Various accuracy distributions were observed in different attacks and noise on both the MNIST and CIFAR-10 datasets. The common pattern of all conditions was that the peaks were NOT located at the zero-noise points. The peaks indicated an optimized defense effect, and the training noise and input noise values indicated the corresponding amplitudes of training noise and input noise.

From the experiment results, we could get a varying range of training noise and running noise which included all peak accuracy of various conditions. We suggest that the choice of noise levels could be inside this varying range.

### 4.3. Robustness

In general, the success of a neural network usually requires a lot of clean data without noise. Noisy data tends to reduce the accuracy of neural networks. Our work showed that adding noise to the training data and input data improved the robustness of the neural networks.

For the MNIST dataset, the effect was not as prominent compared to the CIFAR-10 dataset because those two datasets include different types of images. However, the accuracy did increase after adding noise in most cases, which also enhanced the robustness of corresponding models compared to without adding training noise. In addition, maximum accuracy was also achieved at a certain noise amplitude.

For the CIFAR-10 dataset, after adding noise, the accuracy was in most cases higher for noise below the 100% amplitude level than without adding noise. Even at 80% of the noise amplitude level, the neural network still had a good recognition rate even though the human vision was almost unable to recognize it.

By adding noise during training, the accuracy of the neural network will increase regardless of whether it was subjected to adversarial attacks. It showed that noise had a good effect on improving the robustness of the model.

Although we only conducted experiments on two classic image datasets, the MNIST dataset and the CIFAR-10 dataset, the NFM defended against attacks without knowing the details of the relevant neural network models and datasets. In other words, the NFM has the potential to defend against other 'unknown' deep neural networks and datasets that

may include larger and deeper neural networks and more complex datasets. Furthermore, the NFM makes no restrictions and assumptions about input data and should generalize for input other than images, such as audio. Of course, the final defense effect needs further experiments to verify.

## 5. Conclusions

Without knowing any details about adversarial attacks or corresponding deep neural networks, by adding noise to the model input at the running time and adding noise to the training data at training time, the Noise-Fusion Method (NFM) successfully defended against common adversarial attacks, e.g., the Fast Gradient Signed Method, the Projected Gradient Descent and the Sparse L1 Descent against corresponding deep neural networks on the MNIST and CIFAR-10 datasets. The NFM also improved the robustness of the corresponding deep neural networks.

**Author Contributions:** Methodology and supervision, L.S.; validation and data curation, T.L.; writing original draft preparation, T.L.; writing, review and editing, L.S. and J.H. All authors have read and agreed to the published version of the manuscript.

**Funding:** National Natural Science Foundation of China, funding number 82160347.

**Conflicts of Interest:** The authors declare no conflict of interest.

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
