# Peer review of "Defending Adversarial Attacks against DNN Image Classification Models by a Noise-Fusion Method"

_electronics, doi:10.3390/electronics11121814_

Round 1
Reviewer 1 Report
This paper addresses a noise-based defense method for adversarial attacks against deep neural network models with images as input. The principle used is to add noise to the model input at the running time and the training data at training time.
I suggest changing the title to clarify the contribution domain: “Defending adversarial attacks against DNN image classification models by a noise-fusion method.”
As there are many English grammar issues, the content is difficult to understand. So the paper needs extensive editing. There are also formatting issues such as image and title disposition in figures.
Although, in general, the paper indicates some possibilities of using noise in training and utilization phases, it is not explainable. There are many combinations of the two fusion methods, and it is not easy to calibrate a choice of noise levels for a given application. The authors should indicate how to parameterize both noise levels and adapt them for different image datasets.
The observed differences from applying the proposed method NFM on the MINST and CIFAR-10 datasets merit analysis and explanation. They indicate that the NFM does not present the ability to scale to other neural networks and datasets, as concluded by the paper. There is something different from one dataset to the other that the article failed to address and explain.
The paper's conclusion section is too terse, and the authors should extend it.
Author Response
>>I suggest changing the title to clarify the contribution domain: “Defending adversarial attacks against DNN image classification models by a noise-fusion method.”
We accepted the suggested title: "Defending adversarial attacks against DNN image classification models by a noise-fusion method" and modified the title of paper.
>>As there are many English grammar issues, the content is difficult to understand. So the paper needs extensive editing. There are also formatting issues such as image and title disposition in figures.
1. We corrected English grammar issues we had found in the paper.
2. We corrected the formatting issues of image and title disposition in all figures.
>>Although, in general, the paper indicates some possibilities of using noise in training and utilization phases, it is not explainable. There are many combinations of the two fusion methods, and it is not easy to calibrate a choice of noise levels for a given application. The authors should indicate how to parameterize both noise levels and adapt them for different image datasets.
1. We added a suggestion about the choice of noise levels. From the experiment results, we got a varying range of training noise and running nnoise which included all peak accuracy of various conditions. We added one paragraph into the discussion section 4.2 Accuracy.
2. There was only ONE fusion method which included two steps, adding noise at training time and adding noise at running time. Adding noise at running time was in order to fuse the attack data into noise and adding noise at training time was in order to improve the robustness of models to the noise which added at the running time. Both two steps were needed to defend the adversarial attacks. So there was only one combination case which included both steps.
3. The peak values in panels A-I in Figure 2 and Figure 5 were the optimized noise amplitude values of training and running time noises of two datasets. We modified the section 4.2 Accuracy to indicate the optimized noise amplitudes.
4. One of aspect of our method was "No need to know any details about attacks or models". So we did NOT provide any mechanism of adaptation to specific datasets in the defense method.
>>The observed differences from applying the proposed method NFM on the MINST and CIFAR-10 datasets merit analysis and explanation. They indicate that the NFM does not present the ability to scale to other neural networks and datasets, as concluded by the paper. There is something different from one dataset to the other that the article failed to address and explain.
1. We discussed this in discussion section 4.3 Robustness. NFM defended againest various adversarial attacks on various datasets without need to know any details about attacks or models. NFM already showed the ability to "unknown attacks and unknown models". From the view of NFM, any attacks were "noise". Fusing the noise meant attacks were defended. We added one paragraph into discussion section 4.1 Adversarial Defense Effects.
2. The results did show different defense effects between the MINST and CIFAR-10 datasets while the accuracy showed that defense effects improved on both datasets compared with the case without using NFM defense. Those differences originated from the content of image dataset. The MINST dataset included digital numbers and the CIFAR-10 included images of airplane, automobile, bird, cat, deer, dog, frog, horse, ship, and truck. Other differences also existed among various distributions of noise. All those differences suggested that various defense effects among different conditions while adding noise did improved the accuracy of models in both MINST and CIFAR-10 datasets and various types of noise. In other words, NFM DID defend against attacks although the effects were different at various conditions. We modified discussions section 4.2 Accuracy in the paper.
3. The defense method did improve the correct rate when the attacks added into the models although the correct rates depended on the amplitude of noise.
>>The paper's conclusion section is too terse, and the authors should extend it.
We modified and extend the conclusion section as the request.
Reviewer 2 Report
Manuscript title: Defending adversarial attacks by a noise-fusion method.
Adversarial attack appears due to small disturbance on the image, which is not noticeable to the human eye. To counter the adversarial attack, the author used noise fusion method (NFM) which adds noise to the model inputs and also into the training data. The author used three types of noise: uniform, Gaussian and the Poisson noise in the MNIST and CIFAR-10 data. NFM showed better defense than adversarial training in the CIFAR-10 dataset, but was worse in the MNIST dataset. The results in MNIST and CIFAR-10 are in contrast and so the defending adversarial attacks by the noise-fusion method are inconclusive.
Author Response
>>NFM showed better defense than adversarial training in the CIFAR-10 dataset, but was worse in the MNIST dataset. The results in MNIST and CIFAR-10 are in contrast and so the defending adversarial attacks by the noise-fusion method are inconclusive.
1. NFM showed defense effects in both MNIST and CIFAR-10 datasets although the amplitudes of effects were different in the two datasets. NFM provided better correct rates than the cases without NFM on both datasets. Those differences originated from the content of image dataset. The MINST dataset included digital numbers and the CIFAR-10 included images of airplane, automobile, bird, cat, deer, dog, frog, horse, ship, and truck. We added discussions about the differences in discussion section of the paper.
2. The results in MNIST and CIFAR-10 were NOT in contrast because the results showed improvement both in MNIST and CIFAR-10 although the amplitudes of improvement were different on those two datasets.
Round 2
Reviewer 1 Report
I thank the authors for considering my suggestions and questions during the first review round.
I have appreciated that the paper now clarifies the issues that worried me and that the clarifications qualify the paper for publication. Notwithstanding, I still think that the paper needs grammatical revision to proceed with corrections in content such as:
- the abstract has to be mainly in the present tense;
- No need to know any details about attacks or models,
- as well as more advanced fields while new problems appeared.
- The adversarial attacks impressed us
- causing wrong classification through adding imperceptible small and well-designed attacking data to model input
I cite these sentences that are found on page 1, but the same happens all over the paper. As well as it is important to communicate ideas in the correct language, it is an essential factor for the perception of the Electronics Jornal quality by its readers. Authors are so responsible for a good presentation of their accepted papers.
Author Response
We thank reviewer's nice comments.
>I have appreciated that the paper now clarifies the issues that worried me and that the clarifications qualify the paper for publication. Notwithstanding, I still think that the paper needs grammatical revision to proceed with corrections in content such as:
>- the abstract has to be mainly in the present tense;
We modified the abstract and used mainly in the present tense.
>- No need to know any details about attacks or models,
We modified those sentences by 'Without knowing any details about attacks or models,'.
- as well as more advanced fields while new problems appeared.
We deleted this sentence.
- The adversarial attacks impressed us
We modified this sentence with 'The adversarial attacks cause serious problems'.
- causing wrong classification through adding imperceptible small and well-designed attacking data to model input
We modified this sentence with 'The adversarial attacks cause serious problems by adding imperceptible small and well-designed attacking data to model input,'.
>I cite these sentences that are found on page 1, but the same happens all over the paper. As well as it is important to communicate ideas in the correct language, it is an essential factor for the perception of the Electronics Jornal quality by its readers. Authors are so responsible for a good presentation of their accepted papers.
We are sorry for our poor English. We tried to correct language problems.
By the way, we combined Table 2 and Table 3 together.
Thank you very much.
Reviewer 2 Report
The answers to comments are satisfactory.
Author Response
Thank you very much.